# Effect of Coach Encouragement on the Psychophysiological and Performance Responses of Young Tennis Players

**DOI:** 10.3390/ijerph16183467

**Published:** 2019-09-18

**Authors:** Bulent Kilit, Ersan Arslan, Firat Akca, Dicle Aras, Yusuf Soylu, Filipe Manuel Clemente, Pantelis Theodoros Nikolaidis, Thomas Rosemann, Beat Knechtle

**Affiliations:** 1School of Physical Education and Sports, Namik Kemal University, 59000 Tekirdağ, Turkey; bulentkilit@hotmail.com; 2School of Physical Education and Sports, Siirt University, 56100 Siirt, Turkey; ersanarslan1980@hotmail.com (E.A.); soylusyusuf@gmail.com (Y.S.); 3Faculty of Sport Sciences, Ankara University, 06830 Ankara, Turkey; firatakca@gmail.com (F.A.); diclearasx@gmail.com (D.A.); 4School of Sport and Leisure, Polytechnic Institute of Viana do Castelo, 4960-320 Melgaço, Portugal; filipe.clemente5@gmail.com; 5Instituto de Telecomunicações, Delegação da Covilhã, 6200-001 Covilhã, Portugal; 6Exercise Physiology Laboratory, 18450 Nikaia, Greece; pademil@hotmail.com; 7Institute of Primary Care, University of Zurich, 8091 Zurich, Switzerland; thomas.rosemann@usz.ch; 8Medbase St. Gallen Am Vadianplatz, 9001 St. Gallen, Switzerland

**Keywords:** game-based training, psychophysiological responses, coach encouragement, tennis, on-court tennis drills

## Abstract

This study aimed to compare the effects of coach encouragement during the on-court tennis training drills (OTDs) on the psychophysiological and performance responses of young tennis players. Twenty-five young male tennis players (14.0 ± 0.3 years of age) performed six bouts of each of the four common OTDs; Star, Suicide, Box and Big X (30:60 s; 1:2 work to rest ratio). The heart rate (HR) and total distance covered were monitored using two portable multivariable integrated 10 Hz GPS monitoring devices during all OTDs, and the rating of perceived exertion (RPE-10) and short form Physical Activity Enjoyment Scale (PACES) values were determined after each OTDs bout. The results demonstrated that OTDs with coach encouragement induced significantly higher psychophysiological and performance responses compared to OTDs without coach encouragement (*p* < 0.05). The results of this study confirm that coach encouragement improves the intensity, performance and physical enjoyment level during OTDs. These findings might inform training practices in youth tennis players in order to improve tennis performance.

## 1. Introduction

Tennis is a game characterized by an intermittent high-intensity exercise that includes running at different speeds and making movements as turns, changeovers, strokes, sliding and upper arm involvement [1,2,3]. Considering the metabolic profile, it can be considered an anaerobic sport with aerobic breaks between the rallies over a prolonged match period [1,2,3]. From a physiological perspective, previous studies have shown an average heart rate of approximately 160 beats·min^−1^ (ranging from 120 to 188) in young players [3,4,5]. Simulated tennis matches have shown that young tennis players cover a distance of 2.7–3.4 km and are involved in high-intensity activity (>18 km h^−1^) for approximately 10–25% of the total distance covered [5,6]. Other match characteristics, such as rally duration (~8 s), effective playing time (~22%) and resting time between rallies (~18 s) are also reported in literature [1,4,7]. These results have helped sports scientists and practitioners to prepare tennis-specific training drills and to evaluate the intensity of simulated match-play in young tennis players.

Tennis-specific training drills, or on-court drills (OTDs), involve the actual movement types and patterns used in tennis [7,8]. OTDs are time-efficient because they simultaneously develop physical fitness, technical skill, and tactical awareness [9,10]. Moreover, it is thought that these types of training drills are more motivating and enjoyable compared to off-court training interventions [11,12]. Many studies have confirmed the positive training effect of on-court drills among young tennis players in improving their game-based technical ability, which is directly related to match performance [7,9,10]. Consequently, this game-based training method appears to be a better alternative to traditional options in improving performance responses which are related to tennis match performance in tennis training.

Many previous studies have shown that performance is affected by physical, technical and tactical factors [8,13]. Although these are not the only relevant factors, previous studies have observed that coach encouragement (CE) is a key factor for training and exercise motivation especially in young players [14,15,16]. For example, in a study of the effects of CE during game-based soccer training, Rampinini et al. [17] showed that physiological responses such as heart rate, blood lactate levels, concentration, and rating of perceived exertion were all significantly higher during exercises with CE compared to those where no encouragement was offered. Selmi et al. [16] also found significant positive effects of CE on young soccer players’ heart rate (HR), rating of perceived exertion (RPE) and levels of enjoyment responses in game-based training.

While many studies have examined the effects of CE in team sports, especially soccer [14,15,16,17,18] no study so far has investigated the effects of CE on the psychophysiological and performance responses of tennis players. Such analysis may help to understand the impact of coaches’ encouragement on the player’s intensity, thus being an interesting approach to improve the coaching process and determine encouragement as one more task condition to add in adjusting the practice conditions [19]. Therefore, this study aimed to compare the effects of CE during OTDs on the psychophysiological and performance responses of young tennis players and to analyze which model of training is more effective in improving tennis training performance responses. Based on previous studies, which have indicated an increase in psychophysiological and performance responses with CE, it is hypothesized that the presence of CE would result in an improvement in both the psychophysiological and performance responses of young tennis players.

## 2. Materials and Methods

### 2.1. Participants

Twenty-five right-handed nationally ranked young male tennis players (age: 14.0 ± 0.3 years; training experience 3.4 ± 1.1 years; height: 161.1 ± 7.1 cm; weight: 48.7 ± 6.4 kg; VO_2max_: 52.1 mL·min^−1^·kg^−1^; HR_max_: 201.7 ± 5.4 beats·min^−1^) voluntarily participated in this study prior to the junior tennis season. All the players were accustomed to a training workload of >3 training units per week and had been involved in tennis training and matches for at least 2 years. All the players and their parents were notified of the research procedures, requirements, benefits, and risks before the informed consent forms were provided. The written informed consent forms were obtained from all the subjects and their parents. This study was approved by the Research Ethics Committee of the Hitit University (2018-06/23.01.2018) and was conducted in accordance with the Declaration of Helsinki.

### 2.2. Design and Procedures

A 2-week training period was used to familiarize participants with the testing procedures and OTDs formats. At the end of the familiarization period, players underwent the HTT (Hit and Turn Tennis Test). The study proper was carried out over a 2-week pre-season training period, during which the twenty-five young tennis players who participated were not involved in any other training or matches. After conducting anthropometric measurements, the HTT test was carried out on the players in order to determine the heart rate max (HR_max_) for each player. Then, players performed different OTDs (Star, Suicide, Box and Big X) with and without CE. Each OTDs session was separated by at least 48 h; sessions started with a standardized warm-up lasting for 5 min and consisting of low-intensity running, stretching, short sub-maximum sprints, and integrating tennis-specific actions. During the OTDs, heart rate (HR) responses, total distance covered and the rating of perceived exertion (RPE-10) were determined for each bout of OTDs. Physical Activity Enjoyment Scale (PACES) values were determined after each OTDs bout. Each player was individually filmed, and match characteristics were monitored for the entire duration of the drills. The HTT and OTDs were performed on indoor hard courts at the same time of the day (between 17:00 and 19:00) to ensure similar chronobiological characteristics [20]. The mean temperature during data collection was 15–20 °C and the relative humidity was 40–45%.

### 2.3. Measures

#### 2.3.1. Hit and Turn Tennis Test

The Hit and Turn Tennis Test (HTT) was developed as an acoustically controlled progressive on-court fitness test for tennis players which can be performed simultaneously by one or more players [21]. The test took place in an indoor hard court; it involved specific movements along the baseline (i.e., side steps and running), combined with forehand and backhand stroke simulations at the doubles court corner (distance exactly 11 m). At the beginning of each test level, the player stands in the middle of the baseline with their racket facing forward. Upon hearing a signal, the player turns sideways and runs to the prescribed (i.e., by the CD player) backhand or forehand corner. After making their shot, they return to the middle of the court using side steps or crossover steps while looking at the net. When passing the middle of the baseline again, they turn sideways and continue to run to the opponent’s opposite corner. The test is considered to have ended when players fail to reach the corners in time or are no longer able to carry out the specific movement pattern. The final completed level was used for the determination of tennis-specific endurance capacity and this level was recorded to measure maximum oxygen uptake and peak velocity. Each player’s HR was measured and stored using heart rate monitors (Polar V800, Polar Inc., Finland). The highest HR measurement during the test was recorded as HR_max_. After the test, estimated VO_2max_ was calculated for under-14 boys using the following formula [21]:

VO_2max_ = 30.0 + 1.66 × (player finishes level in HTT).
(1)

#### 2.3.2. On-Court Tennis Training Drills

The OTDs used—Star, Suicide, Box and Big X—were adapted from the study of Reid et al. [7]. These popular tennis-specific drills were selected because they are known to feature in training programs for tennis players in a range of age groups [3,7,9,22]. In addition, these on-court drills show more similarity with tennis match-play in terms of physiological responses elicited [7]. OTDs were randomly played with and without CE at the two-day intervals, while ranking and timing of the players were fixed for tennis-specific drills in order to avoid the chronobiological effect. Players completed six bouts with 30 s of work and 60 s of rest (1:2 work to rest ratio). They were asked to put in maximum effort during the drills. Standardized commandment or positive encouragements (e.g., ‘quick’, ‘good shot’, ‘get back in position quickly’, ‘turn quickly’) were given loudly by the same coach who was near the resting area in the court, and the frequency of verbal feedback was every 5 s during the drills with CE. However, coaches did not give any feedback to players during the drills without CE. Each player’s HR was measured at 1 s intervals throughout the drill and was stored using HR monitors (Polar V800, Polar Inc., Kempele, Finland). The average HR during the drill was calculated by Polar Precision Performance Software TM (PPP4, Kempele, Finland) for each bout. The %HR_max_ responses were also calculated for each bout of OTD according to the following formula: %HR_max_ = (HR in bout/HR_max_ from the HTT) × 100.

Portable multivariable monitoring devices (Bioharness 3, GPS Sports Systems Ltd., Annapolis MD, USA) [23] with integrated 10 Hz GPS units (BT-Q818XT, QStarz, Taipei, Taiwan) [24] were used to record each player’s total distance covered during the drills [21]. The velocity accuracy of the BT-Q818XT is 0.1 m·s^−1^ and the horizontal accuracy of the device is <3.0 m. The RPE-10 rating of the perceived exertion rating scale was presented to each player immediately after each bout of the OTDs. All players were familiarized with the CR-10 scale [25] in the OTDs. This scale has previously been used as an indicator of training intensity in intermittent OTDs bouts [7,26]. The scores were provided individually to avoid hearing other scores, thus ensuring a great accuracy of the answers. An analogue scale was used to help players to visualize the scale.

Enjoyment of physical activity was measured using the short form of the Physical Activity Enjoyment Scale (PACES) after each drill session [27]. This scale, which includes five items scored on a 1–7 Likert scale, has been validated as an indicator of enjoyment level in training in young Turkish players [28]. The scores were provided individually, aiming to decrease the possibilities to be influenced by other answers.

Each player was individually videotaped using two video cameras (60 frames per second) (Sony HDR-CX240 Full HD, Tokyo, Japan) in order to calculate shot numbers in the drills; the analysis of all of the drills was performed by the same experienced researcher. The observer was tested for his reliability level using a test-retest protocol consisting of analyzing and coding the parameters twice, interspaced by 20 days. The intra-class correlation test revealed a value of 0.98, suggesting an excellent reliability level.

### 2.4. Statistical Analyses

Data are expressed as mean ± standard deviation (SD). Before using parametric tests, the assumption of normality was confirmed using the Kolmogorov-Smirnov test. A two-way analyses of variance (ANOVA) with repeated measures was used to determine the effect of the condition (with or without CE), the effect of bouts (six bouts) and their interaction (condition × time) on HR, %HRmax and level of RPE. A paired *t*-test was performed on each dependent variable, including psychophysiological and performance responses, to compare differences between with and without CE conditions for all tennis training drills (Star, Suicide, Box and Big X). When a significant interaction effect was found, Bonferroni post-hoc test was applied in the analysis. The effect size (ES) was calculated for each dependent variable. The thresholds for effect size statistics were as follows [29]: (0.0–0.2) = trivial; (0.2–0.6) = small; (0.6–1.2) = moderate; (1.2–2.0) = large; and > 2.0, very large. Statistical analyses were performed using the SPSS version 17.0 (SPSS, Inc., Chicago, IL, USA) software. The level of statistical significance was set at *p* < 0.05.

## 3. Results

### 3.1. Psychophysiological Responses

Table 1 shows the comparison of the OTDs with and without CE and the bouts played in terms of HR. There was a significant interaction effect on HR (*F*: 3.713; *p* < 0.05 and *η*^2^ = 0.17), main effect for the condition (*F*: 26.221; *p* < 0.05 and *η*^2^ = 0.67) and main effect for the bouts (*F*: 35.001; *p* < 0.05 and *η*^2^ = 0.71). Post hoc analysis showed that HR responses during Star and Suicide drills with and without CE were higher than those during the Box and Big X drills with and without CE. Table 1 also shows that HR responses during the first bout were significantly lower compared to the other five bouts.

The comparison of the OTDs with and without CE and the bouts played in terms of %HR_max_ is shown in Table 2. There was a significant interaction effect on %HR_max_ (*F*: 3.701; *p* < 0.05 and *η*^2^ = 0.17), main effect for the condition (*F*: 26.292; *p* < 0.05 and *η*^2^ = 0.66) and main effect for the bouts (*F*: 35.088; *p* < 0.05 and *η*^2^ = 0.71). Post hoc analysis showed that %HR_max_ responses during Star and Suicide drills with and without CE were higher than those during the Box and Big X drills with and without CE. In addition, the %HR_max_ responses during the first bout were significantly lower compared to the other five bouts, as shown in Table 2.

Table 3 shows the comparison of the OTDs with and without CE and the bouts played in terms of RPE. There was a significant interaction effect on RPE (*F*: 2.651; *p* < 0.05 and *η*^2^ = 0.13), main effect for the condition (*F*: 24.174; *p* < 0.05 and *η*^2^ = 0.52) and main effect for the bouts (*F*: 32.112; *p* < 0.05 and *η*^2^ = 0.66). Post hoc analysis showed that RPE responses during Suicide and Big X drills with and without CE were higher than those during the Star and Box drills with and without CE. In addition, RPE responses to the first bout were significantly lower compared to the other five bouts (see Table 3).

### 3.2. Performance Responses

There was a significant interaction effect on total covered distance (*F*: 3.502; *p* < 0.05 and *η*^2^ = 0.14), main effect for the condition (*F*: 25.453; *p* < 0.05 and *η*^2^ = 0.60) and main effect for the bouts (*F*: 32.332; *p* < 0.05 and *η*^2^ = 0.68). Post hoc analysis showed that total covered distance during the Suicide and Big X drills with and without CE were higher than those during the Star and Box drills with and without CE. A significant interaction effect was found on number of shots played (*F*: 3.842; *p* < 0.05 and *η*^2^ = 0.19), main effect for the condition (*F*: 27.784; *p* < 0.05 and *η*^2^ = 0.72) and main effect for the bouts (*F*: 38.443; *p* < 0.05 and *η*^2^ = 0.79). Post hoc analysis showed that the number of shots played during the Star and Box drills with and without CE was higher than the number played during the Suicide and Big X drills with and without CE.

The average psychophysiological and performance responses of the participants in the six bouts of each OTDs with and without CE are shown in Table 4. The lowest HR and %HR_max_ responses were found in the Box drill with and without CE (173.9 ± 9.6 (86.2%) vs. 171.8 ± 10.1 (85.2%), respectively), whereas the Suicide drill resulted in the highest HR and %HR_max_ responses (178.4 ± 8.5 (88.5%) vs. 176.2 ± 8.8 (87.3%), respectively). The lowest RPE responses (6.4 ± 0.5 vs. 6.8 ± 0.5) were found for the Box drill, whereas the highest RPE responses (7.6 ± 0.5 vs. 7.9 ± 0.5) were found for the Suicide drill with and without CE. The lowest total covered distances were found in the Box drill with and without CE (28.3 ± 3.7 m vs. 26.1 ± 4.0 m, respectively), whereas the Suicide drill resulted in the highest total covered distances (46.8 ± 2.8 m vs. 43.1 ± 2.6 m, respectively). The lowest number of shots (2.9 ± 0.1 vs. 2.7 ± 0.1) were played during the Suicide drill, whereas the highest number of shots (9.7 ± 1.7 vs. 8.9 ± 2.0) were played in the Box drill with and without CE.

## 4. Discussion

This study aimed to examine the effects of CE during OTDs on psychophysiological and performance responses of young tennis players. To the best of our knowledge, the present study is the first to undertake such a thorough examination of these variables in young tennis players. One of the main findings of the study is that players had higher physiological responses (HR, %HR_max_ and RPE levels) in OTDs when CE was provided. The results from the present study are consistent with many studies of small sided games (SSGs) in soccer, which supports the hypothesis that CE helps increase psychophysiological responses in game-based training drills [14,15,16,17,18]; when coaches offer encouragement, players tend to try harder.

In this study, exercise intensity was determined by monitoring HR, %HR_max_ and RPE levels during the OTDs. The most popular and easy method of measuring intensity of exercise is HR monitoring, although it has some limitations, especially in games with short duration and high intensity [30]. This study found that average %HR_max_ responses to OTDs ranged from 88.0 to 90.7%. This is consistent with the findings of Reid et al. [7] who reported %HR_max_ responses ranging from % 90.2–92.0 for four common OTDs; it can therefore be seen that our findings are lower than those of previous studies. In addition to these results, young tennis players had greater HR and %HR_max_ responses in OTDs with CE compared to those where there was no CE. Several studies have previously shown that CE increases game or exercise intensity, especially in young players [15,16,18]. For example, Rampinini et al. [17] showed that CE (compared to lack of CE) induced higher %HR_max_ responses during four-a-side SSGs. Another study showed the effects of CE in increasing %HR_max_ responses in young players involved in four-a-side SSGs [16].

Recently, studies have suggested that different measurement methods should be used in combination with HR, such as RPE, which is considered to be a viable method for tracking internal loads using low cost, easily accessible procedures [13,31]. This study found that, similarly to HR responses, verbal CE also affected RPE levels; young players had greater RPE responses in OTDs with CE compared to those without CE. Furthermore, our RPE findings showed higher and different values from a previous study, where RPE responses were found to range from 5.0–7.6 for all OTDs [7]. Such differences may be explained by the participants’ age, level and the demands of the sport. Surprisingly, other studies have shown higher physiological responses such as HR and RPE in the game-based training drills with CE, without finding statistically significant differences [14,18]. These differences may be explained by the type (mild vs. strong) and the frequency of CE.

Apart from these important objective and subjective measurements, studies have recently increasingly reported the use of other popular scales in sports, such as PACES, which assesses the levels of enjoyment relative to a given activity [12,16]. The findings of this study that the OTDs with CE groups for all OTDs types had higher physical enjoyment responses (Figure 1) and lower perceived exertion compared to OTDs without CE are consistent with studies of game-based high intensity training drills [11,16]. The possible explanation for such outcomes might be that OTDs which involve the actual movement types and patterns of racket use found in tennis [7,8] are more motivating and enjoyable compared to off-court training methods. Many studies have supported the positive effect of motivation and enjoyment, which is directly related with effort expenditure and participation in sport, particularly in younger players [31,32,33].

Determination of time-motion characteristics and performance responses in OTDs is as important as determination of psychophysiological responses to tennis drills. The recent development of technological tools such as wearable and multivariable monitoring devices including 10–15 Hz GPS gives more reliable results than video-based analysis [34]. These tools offer a highly practical way of monitoring time-motion characteristics of players such as total distance and distance covered at different speeds during tennis training and matches [6,35,36]. In terms of CE, recent studies have found that using verbal CE also has an effect on time-motion characteristics and performance responses in game-based training [7,14,37]. Therefore, CE is an important factor which should not be disregarded. However, few studies have focused on time-motion characteristics and performance responses in game-based training in tennis players [3,7]. Taken together, our results show that CE affects not only the psychophysiological responses but also time-motion characteristics and performance responses. We found that players covered greater total distance and played a greater number of shots in OTDs with CE.

There are some limitations that need to be acknowledged and addressed regarding the present study. The study is limited to a relatively small sample size of young male tennis players. Another limitation is different court surface (clay or grass) and court conditions (indoor or outdoor) affecting the level of psychophysiological and performance responses of players. A major strength of this study is the use of the frequency of verbal encouragement, as this allowed us to perform standardized measurements in on-court conditions during drills. Further research is required to investigate the effect of different forms of coach encouragement on psychophysiological and performance responses of on-court tennis training drills in different courts.

As practical implications, we should to consider that the majority of children at this age spend most of their time at school and have little spare time outside of their studies. This means that they cannot spend much time on tennis training. Therefore, game-based training methods seem a better alternative in order to improve performance responses in tennis training which is related to tennis match performance. Moreover, it is seen that these types of training drills with coach encouragement are more motivating and enjoyable compared to off-court training interventions. It is believed that planning of training programs with consideration of match-play conditions will be more effective for both trainers and players.

## 5. Conclusions

Many studies have shown the effects of CE in team sports, especially soccer. In contrast, no study has investigated the effects of CE on training and exercise performance in young tennis players. The results of this study indicate that coach encouragement improves intensity, performance and physical enjoyment levels during on-court tennis training drills. These findings seem significant in terms of improving the tennis performance of young tennis players.

## Figures and Tables

**Figure 1 ijerph-16-03467-f001:**
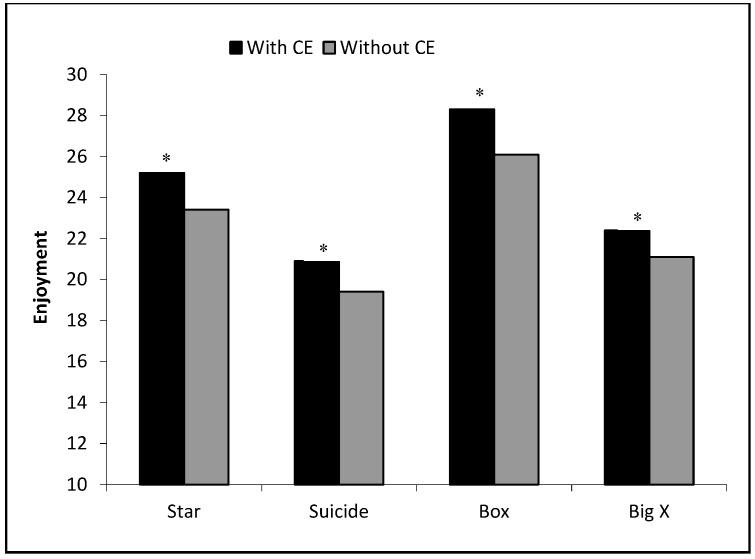
Enjoyment scale values for the with and without CE on-court tennis drills. * Significant differences from without CE. *p* < 0.05.

**Table 1 ijerph-16-03467-t001:** Heart rate (beats·min^−1^) values (mean ± SD) for the on-court tennis drills in each bout.

Number of Bouts	Star		Suicide		Box		Big X	
	With CE	Without CE	With CE	Without CE	With CE	Without CE	With CE	Without CE
Bout-1	169.4 ± 9.2 *	166.7 ± 10.5 *	172.1 ± 7.4 *	170.3 ± 6.6 *	166.5 ± 10.1 *	164.6 ± 10.5 *	167.7 ± 10.6 *	165.8 ± 10.2 *
Bout-2	175.9 ± 9.7	173.3 ± 9.7	176.9 ± 6.9	174.8 ± 6.5	172.0 ± 9.8	171.2 ± 10.3	173.5 ± 9.5	171.9 ± 10.0
Bout-3	176.6 ± 8.9	173.9 ± 9.5	178.0 ± 7.2	175.8 ± 6.8	174.4 ± 9.7	172.4 ± 9.8	174.0 ± 9.2	172.8 ± 9.7
Bout-4	177.8 ± 9.2	175.3 ± 9.8	179.4 ± 6.1	176.5 ± 7.1	176.0 ± 9.2	173.9 ± 9.1	175.4 ± 9.3	173.7 ± 9.5
Bout-5	178.5 ± 9.0	176.9 ± 9.1	181.1 ± 5.4	179.4 ± 5.7	176.9 ± 9.0	173.1 ± 9.4	176.5 ± 9.1	174.3 ± 9.6
Bout-6	179.3 ± 8.4	177.4 ± 10.7	183.0 ± 5.0	180.2 ± 6.4	177.5 ± 9.3	175.4 ± 10.7	178.1 ± 9.4	175.9 ± 10.2

* Significant difference from the other five bouts, *p* < 0.05. CE = coach encouragement.

**Table 2 ijerph-16-03467-t002:** Percentage of maximum heart rate (%HR_max_) values (mean ± SD) for the on-court tennis drills in each bout.

Number of Bouts	Star		Suicide		Box		Big X	
	With CE	Without CE	With CE	Without CE	With CE	Without CE	With CE	Without CE
Bout-1	84.2 ± 4.5 *	82.6 ± 5.1 *	85.3 ± 3.7 *	84.4 ± 3.4 *	82.5 ± 4.9 *	81.6 ± 5.2 *	83.1 ± 5.3 *	82.2 ± 5.1 *
Bout-2	87.2 ± 4.7	85.9 ± 4.9	87.7 ± 3.5	86.7 ± 3.2	85.3 ± 4.8	84.9 ± 5.1	86.0 ± 4.7	85.2 ± 4.9
Bout-3	87.6 ± 4.4	86.2 ± 4.8	88.2 ± 3.6	87.2 ± 3.5	86.5 ± 4.6	85.5 ± 4.6	86.3 ± 4.6	85.7 ± 4.8
Bout-4	88.2 ± 4.6	86.9 ± 5.0	88.9 ± 3.1	87.5 ± 3.6	87.3 ± 4.2	86.2 ± 4.4	87.0 ± 4.6	86.1 ± 4.6
Bout-5	88.5 ± 4.4	87.7 ± 4.5	89.8 ± 2.8	88.9 ± 3.0	87.7 ± 4.1	85.8 ± 4.6	87.5 ± 4.5	86.4 ± 4.7
Bout-6	88.9 ± 4.2	88.0 ± 5.2	90.7 ± 2.6	89.3 ± 3.3	88.0 ± 4.3	87.0 ± 5.3	88.3 ± 4.6	87.2 ± 5.2

* Significant difference from the other five bouts, *p* < 0.05.

**Table 3 ijerph-16-03467-t003:** Rating of perceived exertion (RPE-10) values (mean ± SD) for the on-court tennis drills in each bout.

Number of Bouts	Star		Suicide		Box		Big X	
	With CE	Without CE	With CE	Without CE	With CE	Without CE	With CE	Without CE
Bout-1	6.5 ± 0.6 *	6.7 ± 0.7 *	7.3 ± 0.5 *	7.6 ± 0.7 *	6.1 ± 0.5 *	6.4 ± 0.5 *	7.1 ± 0.4 *	7.3 ± 0.5 *
Bout-2	6.7 ± 0.7	7.0 ± 0.6	7.6 ± 0.5	7.8 ± 0.7	6.3 ± 0.4	6.7 ± 0.5	7.4 ± 0.5	7.5 ± 0.6
Bout-3	6.9 ± 0.6	7.2 ± 0.6	7.6 ± 0.4	7.8 ± 0.6	6.3 ± 0.5	6.8 ± 0.5	7.4 ± 0.5	7.7 ± 0.6
Bout-4	7.0 ± 0.6	7.2 ± 0.6	7.7 ± 0.5	7.9 ± 0.5	6.4 ± 0.6	6.8 ± 0.4	7.5 ± 0.4	7.8 ± 0.5
Bout-5	7.0 ± 0.6	7.3 ± 0.5	7.8 ± 0.5	8.0 ± 0.6	6.6 ± 0.5	7.0 ± 0.4	7.6 ± 0.4	7.8 ± 0.5
Bout-6	7.1 ± 0.5	7.3 ± 0.4	7.9 ± 0.4	8.2 ± 0.4	6.7 ± 0.4	7.1 ± 0.5	7.7 ± 0.3	7.9 ± 0.4

* Significant difference from the other five bouts, *p* < 0.05.

**Table 4 ijerph-16-03467-t004:** Psychophysiological and performance responses of young tennis players for with CE and without CE.

Variables	Variables	With CE	Without CE	Difference	d	ES Magnitude
Star	HR (beats·min^−1^)	176.3 ± 9.4 *	173.9 ± 10.0	2.4	0.25	Small
	%HR_max_ (%)	87.4 ± 4.1 *	86.2 ± 4.9	1.2	0.27	Small
	RPE	6.9 ± 0.6 *	7.1 ± 0.6	−0.2	0.33	Small
	PACES	25.2 ± 4.4 *	23.4 ± 4.9	1.8	0.39	Small
	Total Distance (m)	28.8 ± 3.4 *	26.9 ± 3.8	1.9	0.53	Small
	Number of Shots	8.8 ± 1.2 *	8.0 ± 1.5	0.8	0.59	Small
Suicide	HR (beats·min^−1^)	178.4 ± 8.5 ^Ω^	176.2 ± 8.8	2.2	0.25	Small
	%HR_max_ (%)	88.5 ± 4.2 ^Ω^	87.3 ± 4.3	1.2	0.28	Small
	RPE	7.6 ± 0.5 ^Ω^	7.9 ± 0.6	−0.3	0.54	Small
	PACES	20.9 ± 6.3 ^Ω^	19.4 ± 6.8	1.5	0.23	Small
	Total Distance (m)	46.8 ± 2.8 ^Ω^	43.1 ± 2.6	3.7	1.37	Large
	Number of Shots	2.9 ± 0.1 ^Ω^	2.7 ± 0.1	0.2	1.99	Large
Box	HR (beats·min^−1^)	173.9 ± 9.6 ^#^	171.8 ± 10.1	2.1	0.21	Small
	%HR_max_ (%)	86.2 ± 4.7 ^#^	85.2 ± 5.0	1.0	0.21	Small
	RPE	6.4 ± 0.5 ^#^	6.8 ± 0.5	−0.4	0.80	Moderate
	PACES	28.3 ± 3.7 ^#^	26.1 ± 4.0	2.2	0.57	Small
	Total Distance (m)	42.5 ± 5.1 ^#^	40.8 ± 5.7	1.7	0.31	Small
	Number of Shots	9.7 ± 1.7 ^#^	8.9 ± 2.0	1.8	0.97	Moderate
Big X	HR (beats·min^−1^)	174.2 ± 9.5 ^¥^	172.4 ± 10.0	1.8	0.18	Trivial
	%HR_max_ (%)	86.4 ± 4.8 ^¥^	85.5 ± 5.1	0.9	0.18	Trivial
	RPE	7.5 ± 0.4 ^¥^	7.7 ± 0.5	−0.2	0.44	Small
	PACES	22.4 ± 5.1 ^¥^	21.1 ± 4.9	1.3	0.26	Small
	Total Distance (m)	42.6 ± 3.1 ^¥^	40.9 ± 3.3	1.7	0.54	Small
	Number of Shots	3.0 ± 0.3 ^¥^	2.9 ± 0.3	0.1	0.54	Small

HR = heart rate; %HR_max_ = percentage of maximum heart rate; RPE = rating of perceived exertion; PACES = Physical Activity Enjoyment Scale. Values are given as mean ± SD. * Significant differences from Star without CE, *p* < 0.05. ^Ω^ Significant differences from Suicide without CE, *p* < 0.05. ^#^ Significant differences from Box without CE, *p* < 0.05. ^¥^ Significant differences from Big X without CE, *p* < 0.05.

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
