# Peer review of "Effect of Coach Encouragement on the Psychophysiological and Performance Responses of Young Tennis Players"

_ijerph, 2019, doi:10.3390/ijerph16183467_

Round 1

Reviewer 1 Report

The paper is well written. Posses scientific and practical significance as well.

I suggest only small corrections:

line 120 - "cones" should be changed on "corners"

line 295 - word "according" was used two times.

The reference section

The Authors should decide which version of title writing would like to use. All words as a capital letter or maybe lowercase. Please check it and correct according with Int.J.Environ.Res. Public Health requirements.

I also strongly suggest citing the paper bellow:

"What do we want and what do we get from the coach? Preferred and perceived leadership in male and female team sports" Human Movement 2019 vol. 20 (3), 38-47, DOI: https://doi.org/10.5114/hm.2019.79734

Author Response

REVIEWER 1

The paper is well written. Posses scientific and practical significance as well.

I suggest only small corrections:

line 120 - "cones" should be changed on "corners"

AUTHORS: cones was changed with corners

line 295 - word "according" was used two times.

AUTHORS:  extra according was deleted

The reference section

The Authors should decide which version of title writing would like to use. All words as a capital letter or maybe lowercase. Please check it and correct according with Int.J.Environ.Res. Public Health requirements.

AUTHORS:  this section was re-organized

I also strongly suggest citing the paper bellow:

"What do we want and what do we get from the coach? Preferred and perceived leadership in male and female team sports" Human Movement 2019 vol. 20 (3), 38-47, DOI: https://doi.org/10.5114/hm.2019.79734

AUTHORS: this reference was added and then references were re-organized in the manuscript

Thank you for your review and helpful comments. We have made substantial changes to the manuscript. We believe that the revised version is improved and hope that we have addressed the shortcomings highlighted.

Reviewer 2 Report

General Comments:This study investigated the impact of coaches’ encouragements on player’s intensity during on-court tennis training drills. Overall, this is a nice study that could have great practical application when integrated with athlete monitoring strategies. The authors are commended on their efforts thus far. The study is well designed and well-written, with a great introduction proposing the usefulness of the topic and a clear outline of the research question. However, the clarity of the results need to be improved to allow this manuscript to be ready for publication. 

Specific Comments:

Introduction-

The authors are commended on their introduction of the topic and framing of the problem and research question.

Methods-

Was the Max heart rate used obtained during the Hit and Turn Tennis Test used as the Maximal HR in all calculations? Was this typical close to the age-predicted heart rate max? It may be important to have a secondary measure to ensure participants were getting to maximal heart rate, if this data is going to be used to predict VO2max; therefore the cardiovascular system was the reason for test termination and not just the lack of locomotion speed (to get to the cone in time) becoming the limiting factor for test termination.

Statistical Analysis-

Line 164: Why was a paired t-test used to compare the PACES score for the star, suicide, Box and Big X tennis training drills? Shouldn’t a two-way Repeated Measure Analysis of Variance (RM ANOVA) be more effect (Condition X Drill), with subsequent post-hoc analyses-similar to the two-way RM ANOVA?

Line 169: need to explain what effect size you are using (I believe eta squared). Why was eta squared used an not Cohen’s D (or Cohen’s F), typically seen in sport science?

Line 172: Other than the two comments above, nice job with the clarity of the statistical approaches.

Line 175: The statistical analysis section suggests the first analysis run would be a t-test, but the results for this are not presented or unclear. Keep the same format in the results as outlined in the statistical analysis section.

Like 182: Formatting is unclear. Make the adjustments to allow data to be on one line and not jump to the next.

For a lot of the statistical difference, there are only a few beats difference in the average heart rate. Would this be practically significant? Would this alter the over intensity of the drill from an energy system stand-point or time to exhaustion?

Line 232: This graph should be revised to include both an X and Y axis on the inside of the graph frame. The over appearance of this graph could be improved. Again there are no units on the graph, but are these significant difference also practically (clinically) significant?

Discussion-

Overall the discussion is well-written and incorporates relevant literature.

Does this study have no limitations?

Author Response

REVIEWER 2

General Comments: This study investigated the impact of coaches’ encouragements on player’s intensity during on-court tennis training drills. Overall, this is a nice study that could have great practical application when integrated with athlete monitoring strategies. The authors are commended on their efforts thus far. The study is well designed and well-written, with a great introduction proposing the usefulness of the topic and a clear outline of the research question. However, the clarity of the results need to be improved to allow this manuscript to be ready for publication. 

AUTHORS: Thank you for your review and helpful comments. We have made substantial changes to the manuscript. We believe that the revised version is improved and hope that we have addressed the shortcomings highlighted.

Specific Comments:

Introduction-

The authors are commended on their introduction of the topic and framing of the problem and research question.

AUTHORS: Thank you for your feedback.

Methods-

Was the Max heart rate used obtained during the Hit and Turn Tennis Test used as the Maximal HR in all calculations? Was this typical close to the age-predicted heart rate max? It may be important to have a secondary measure to ensure participants were getting to maximal heart rate, if this data is going to be used to predict VO2max; therefore the cardiovascular system was the reason for test termination and not just the lack of locomotion speed (to get to the cone in time) becoming the limiting factor for test termination.

AUTHORS: We apologize for the confusion. Yes. The highest HR was recorded as HRmax during the Hit and Turn Tennis Test. Our measured HRmax results during Hit and Turn test was similar compared to age-predicted (220-age) HRmax (202 ± 5 vs 205 ± 1, respectively). It seems that similar physiological load was obtained using the Hit and Turn test. Sport-specific nature of Hit and Turn test (which employed on tennis court and included both stimulation of upper and lower extremity muscles and important tennis-specific correlates such as acceleration, deceleration and strokes with racket) gives some similarity with lab-based tests, hence we did not need secondary measurements (Ferrauti et al,. 2011).

Statistical Analysis-

Line 164: Why was a paired t-test used to compare the PACES score for the star, suicide, Box and Big X tennis training drills? Shouldn’t a two-way Repeated Measure Analysis of Variance (RM ANOVA) be more effect (Condition X Drill), with subsequent post-hoc analyses-similar to the two-way RM ANOVA?

AUTHORS: We apologize for the lack of knowledge. A paired t-test was performed on each dependent variable, including psychophysiological and performance responses, to compare differences between with and without CE conditions for all tennis training drills (Star, Suicide, Box and Big X) (Table 4).

Line 169: need to explain what effect size you are using (I believe eta squared). Why was eta squared used an not Cohen’s D (or Cohen’s F), typically seen in sport science?

AUTHORS: A two-way analyses of variance (ANOVA) with repeated measures was used to determine the effect of the condition (with or without CE), the effect of bouts (6 bouts) and their interaction (condition × time) on HR, %HRmax and level of RPE. When a significant interaction effect was found, Bonferroni post-hoc test was applied in the analysis. Because of this, we used an eta squared values in our results.

Line 172: Other than the two comments above, nice job with the clarity of the statistical approaches.

AUTHORS: Thank you for your evaluation.

Line 175: The statistical analysis section suggests the first analysis run would be a t-test, but the results for this are not presented or unclear. Keep the same format in the results as outlined in the statistical analysis section.

AUTHORS: Statistical analysis section was  re-organized.

Like 182: Formatting is unclear. Make the adjustments to allow data to be on one line and not jump to the next.

AUTHORS: The tables were re-organized.

For a lot of the statistical difference, there are only a few beats difference in the average heart rate. Would this be practically significant? Would this alter the over intensity of the drill from an energy system stand-point or time to exhaustion?

AUTHORS: These tennis-specific drills (Star, Suicide, Box and Big X) were selected because they have similar exercise intensity to the HIIT drills (≥85% of HRmax) (Reid et al., 2008). After this point, we think that a few beats differences in HR (statistically significant) might be very important in practically for distribution of energy system and exhaustion. However, the average HR obtained from games or drills may not be the best variable to evaluate the internal load for players. Along with the HR responses, the other important internal responses such as RPE and LA, should be use as a combined physiological responses in different drills.

Line 232: This graph should be revised to include both an X and Y axis on the inside of the graph frame. The over appearance of this graph could be improved. Again there are no units on the graph, but are these significant difference also practically (clinically) significant?

AUTHORS: This graph was re-organized.

Discussion-

Overall the discussion is well-written and incorporates relevant literature.

Does this study have no limitations?

AUTHORS: Limitations were added

Reviewer 3 Report

Design of the study could be improved. For example the timing of CE vs no CE should be counterbalanced with a randomization procedure to determine the order of treatment.

The type of CE should be more clearly explained.

The tables (particularly table #1) are poorly presented (heading on dif page, mix of bold and non-bold, #s don't line up horizontally, not clear what the number below each measure represents)

Limitations of the study should be noted.

Author Response

REVIEWER 3

Design of the study could be improved. For example the timing of CE vs no CE should be counterbalanced with a randomization procedure to determine the order of treatment.

AUTHORS: We apologize for the confusion. The study design (including selection of the drills, encouragement and ranking and timing of the players) was clearly explained.

The type of CE should be more clearly explained.

AUTHORS: The type of CE was clearly explained.

The tables (particularly table #1) are poorly presented (heading on dif page, mix of bold and non-bold, #s don't line up horizontally, not clear what the number below each measure represents)

AUTHORS: The tables were re-organized

Limitations of the study should be noted.

AUTHORS: Limitations were added

Thank you for your review and helpful comments. We have made substantial changes to the manuscript. We believe that the revised version is improved and hope that we have addressed the shortcomings highlighted.

Round 2

Reviewer 2 Report

The authors did a great job addressing concerns and this manuscript is ready for publication. I would make one following comment, because this is interesting and may even make the work event stronger. 

REVIEWER: For a lot of the statistical difference, there are only a few beats difference in the average heart rate. Would this be practically significant? Would this alter the over intensity of the drill from an energy system stand-point or time to exhaustion?

AUTHORS: These tennis-specific drills (Star, Suicide, Box and Big X) were selected because they have similar exercise intensity to the HIIT drills (≥85% of HRmax) (Reid et al., 2008). After this point, we think that a few beats differences in HR (statistically significant) might be very important in practically for distribution of energy system and exhaustion. However, the average HR obtained from games or drills may not be the best variable to evaluate the internal load for players. Along with the HR responses, the other important internal responses such as RPE and LA, should be use as a combined physiological response in different drills.”

I think this is a great point. With that being said, would it be more valuable to compare Training Impulse values, which could incorporate time spent in varying zones?

Author Response

REVIEWER 2

For a lot of the statistical difference, there are only a few beats difference in the average heart rate. Would this be practically significant? Would this alter the over intensity of the drill from an energy system stand-point or time to exhaustion?

AUTHORS: As you said time spent in varying HR zones is very important in this section to evaluate differences in especially HR responses between training drills, however we did not measure it. Evaluating time spent in varying HR zones might give us more detailed knowledge about the distribution of energy systems and exhaustion during tennis drills. We keep your recommendation in our mind for the future studies.

Reviewer 3 Report

The authors have addressed my previous concerns

Author Response

AUTHORS: No changes are required